# Efficient Synthesis of High-Performance Anion Exchange Membranes by Applying Clickable Tetrakis(dialkylamino)phosphonium Cations

**DOI:** 10.3390/polym15020352

**Published:** 2023-01-09

**Authors:** Yu Wang, Yudong Wang, Sushant Sahu, August A. Gallo, Xiao-Dong Zhou

**Affiliations:** 1Department of Chemistry, University of Louisiana at Lafayette, Lafayette, LA 70504, USA; 2Institute for Materials Research and Innovation, University of Louisiana at Lafayette, Lafayette, LA 70504, USA; 3Department of Chemical Engineering, University of Louisiana at Lafayette, Lafayette, LA 70504, USA

**Keywords:** anion exchange membrane, AEM, tetrakis(dialkylamino)phosphonium, “click” chemistry

## Abstract

Tetrakis(dialkylamino)phosphonium (TKDAAP) compounds exhibit extraordinary base resistance, a prerequisite feature for high-performance anion exchange membranes (AEMs). It is, however, challenging to synthesize a TKDAAP compound with reactive functionality that can be used to link the cation to a polymer backbone. In this study, two TKDAAP compounds with alkyne functionality were synthesized and incorporated into an azide-modified SBS triblock copolymer backbone via Cu(I)-catalyzed alkyne–azide cycloaddition (CuAAC) “click” chemistry. The properties of the resulting AEMs were characterized. It was found that (1) the triazole linker between the cation and the polymer backbone was stable under alkaline conditions; (2) varying the substituents of TKDAAP compounds could dramatically alter the stability; and (3) increasing the hydrophilicity of the AEM was an efficient way to enhance its ionic conductivity. Using clickable TKDAAP compounds makes it easy to combine various cations into polymer backbones with adjustable cation content, thus potentially leading to an efficient way to screen a wide variety of polyelectrolyte structures to identify the most promising candidates for high-performance AEMs.

## 1. Introduction

Anion exchange membrane fuel cells (AEMFCs) and electrolyzers are promising technologies for the clean and efficient generation of renewable power and chemicals [1]. Compared with proton exchange membrane fuel cells (PEMFCs), AEMFCs can be operated without using expensive Pt group metal catalysts at the cathode due to improved oxygen reduction reaction kinetics, thus potentially lowering the cost significantly [1,2]. Anion exchange membranes (AEMs) are used in fuel cells and electrolyzers as the core components for selective ion conduction. The development of high-performance AEMs is critical for the commercialization of these electrochemical devices [3]. High-performance AEMs should have the following requirements: (1) Proper mechanical and physical properties: An AEM should be a thin film that is not too brittle, nor too soft. It should have sufficient tensile strength and modulus. When used in an aqueous system, it should have proper water-absorbing properties and water permeability. However, the membrane should not swell too much when immersed in water. In the wet state, the membrane should still have enough strength. (2) High ionic conductivity: The higher the ionic conductivity is, the better the performance of a fuel cell or electrolyzer is. Using Nafion membranes as a reference, those membranes typically exhibit a high proton conductivity of up to 200 mS cm^−1^ at 80 ${}^{\circ }C$ in water [4]. Thus, an AEM should have a OH^−^ conductivity close to or higher than this reference. (3) Good stability: AEMFCs are operated under strongly alkaline conditions. Many organic compounds change their structures under such conditions. The AEMs used in fuel cells and electrolyzers should maintain their performance after a long time of operation, i.e., in the order of 10,000 h.

In recent years, the reported conductivity values of AEMs have increased dramatically. The conductivity of some commercial products has already reached or exceeded 100 mS cm^−1^ at 60 ${}^{\circ }C$ [5]. However, the majority of reported AEMs are based on quaternary ammonium (QA) ion structures and simple imidazolium (IM) ions [3], most of which are prone to degradation under alkaline conditions. The degradation of cations is the main reason for the loss of conductivity. Additionally, the degradation of the backbone of the AEM would significantly reduce the mechanical property of the membrane and thus cause the failure of the AEMFC. Durability issues associated with AEMs remain a primary challenge in accelerating the commercialization of AEMFCs [6]. To date, no cells have been reported to show sufficient durability during long-term operation, i.e., ∼2000 h, at DOE-relevant current densities (0.6 A cm^−2^), especially at high temperatures and under low-humidity conditions [7]. The durability of AEMs is limited by the intrinsic stability of QA and simple IM cations. The intrinsic stability of organic cations can be evaluated using the standardized protocol developed by Hugar et al. [8]. It was found that even the most alkaline-stable QA cations showed degradation of a few percentage points in 2 M KOH in CD_3_OH at 80 ${}^{\circ }C$ after 30 days (Figure 1). On the contrary, many tetrakis(dialkylamino)phosphonium (TKDAAP) and penta-substituted imidazolium cations showed no observable degradation under identical conditions or even in 5 M KOH [9,10]. It is desirable to go beyond QA and simple IM cations and utilize these more alkaline-stable cations for AEM development. However, in general, these ultra-stable cations have more complex structures. The synthesis of these alkaline-stable cation units and their incorporation into polymers are usually more challenging than the synthesis of QA-based AEMs. Thus, the critical issue is to identify an efficient synthetic method to utilize alkaline-stable cations for more efficient AEM development.

Many TKDAAP cations are extremely base resistant [11,12]; however, the most challenging part of using TKDAAP cations for AEMs is the synthesis of a TKDAAP compound with one reactive functional group that can be used to link to the polymer backbone. To solve this problem, two clickable TKDAAP compounds with alkyne functionality were synthesized using a method similar to that used in a recent report [13]. In general, clickable TKDAAP compounds can be synthesized simply, in two steps, without rigorous purification. Alkyne groups can be used to link the cations to azide-modified polymers via CuAAC “click” chemistry, which results in a nearly quantitative yield with a minimal level of side reactions. Using clickable TKDAAP cations, it becomes possible to design cations and polymer backbones separately and combine them in different ways. The easy synthesis of clickable TKDAAP compounds and the highly efficient coupling reaction between the cation and the polymer backbone could pave the way for more efficient synthesis and evaluation of TKDAAP-based AEMs.

## 2. Materials and Methods

### 2.1. Materials

Benzyl bromide (Alfa Aesar, Haverhill, MA, USA; 99%), sodium azide (NaN_3_; SigmaAldrich, St. Louis, MI, USA; 99.5%), phosphorus(V) chloride (PCl_5_; Alfa Aesar; 98%), morpholine (Thermo Scientific, Waltham, MA, USA; 99%), triethylamine (TEA; Oakwood Chemical, Hampton, SC, USA; 99.5%), propargylamine (Oakwood Chemical, Estill, SC, USA; 98%), potassium hexafluorophosphate (KPF_6_; TCI America, Portland, OR, USA; 95%), iodomethane (Alfa Aesar; 99%), 1,1,4,7,7-pentamethyldiethylenetriamine (PMDETA; Acros Organics, Geel, Belgium; 99%), N-methylcyclohexylamine (Alfa Aesar; 98%), CuBr_2_ (Acros Organics; 99%), N-methylpiperidine (Acros Organics; 99%), propargyl bromide (80 wt% solution in toluene; Acros Organics); polystyrene-block-polybutadiene-block-polystyrene (SBS; average molecular weight of 140 K; Sigma Aldrich), hydrogen bromide (33% *w*/*w* in glacial acetic acid; Alfa Aesar), 18-crown-6 ether (Oakwood Chemical, Estill, SC, USA; 99%), propargyl alcohol (Oakwood Chemical, 99%), 1-hexyne (Alfa Aesar; 98%), 2-trimethylsilylethanol (Oakwood Chemical; 95%), MgSO_4_ (anhydrous; Thermo Scientific; 99.5%), K_2_CO_3_ (Thermo Scientific; 99%); KOH (Pellets; certified ACS; Fisher Chemical, Waltham, MA, USA), CDCl_3_ (0.03% *v*/*v* TMS; Thermo Scientific; 99.8%), CD_3_OH (Acros Organics; 99.5%), and DMSO-$d_{6}$ (Thermo Scientific; 99.5%) were used as received. Cu(0) wire (d=0.25mm; Alfa Aesar; 99.9%) was washed with hydrochloric acid and methanol and dried. Solvents, dimethylformamide (DMF), toluene, chlorobenzene, isopropyl alcohol(*i*-PrOH), dimethyl sulfoxide (DMSO), diethyl ether (Et_2_O), acetone, methanol (MeOH), tetrahydrofuran (THF) of HPLC grade or certified by ACS were purchased from Fisher Scientific.

### 2.2. Characterization

Nuclear magnetic resonance (NMR) spectroscopy was performed using a Bruker Avance AVIII-400 MHz NMR spectrometer or a Varian 400 MHz NMR spectrometer. The ${}^{1}$H NMR spectra were referenced to TMS or residual protio solvents (TMS, 0 ppm; CHCl_3_, 7.26 ppm; CHD_2_OH, 3.31 ppm; DMSO-$d_{5}$, 2.50 ppm). The ${}^{13}$C NMR spectra were referenced to the solvent signal (CDCl_3_, 77.23 ppm; DMSO-$d_{6}$, 39.51 ppm). The ${}^{31}$P NMR spectra were referenced to H_3_PO_4_ (85%) in H_2_O (0 ppm). All spectra were analyzed using MestReNova software.

Fourier transform infrared spectroscopy (FTIR) was performed using an Agilent Cary 630 attenuated total reflection (ATR) FTIR spectrometer.

Mass spectrometry (MS) was conducted using a high-performance liquid chromatography (HPLC) instrument coupled with an accurate mass electrospray ionization (ESI) mass spectrometer; specifically, we used an Agilent 1260 Infinity II Quaternary liquid chromatography instrument coupled with an Agilent 6230 Electrospray Time-of-Flight mass spectrometer. The samples were run in positive mode ionization with a capillary voltage of 4000 V. The drying gas (nitrogen) temperature was 325 ${}^{\circ }C$, delivered at 10 L/min, and the fragmentor voltage was set to 150 V. Only flow-through injection was utilized (direct injection from LC instrument to mass spectrometer). The mobile phases used were as follows: A—30% LCMS-grade water with 0.1% formic acid; B—70% LCMS-grade acetonitrile with 0.1% formic acid at a flow rate of 0.4 mL/min.

Electrochemical analysis: The ionic conductivity of the AEMs was measured using a Biologic VMP3 multi-channel potentiostat (Bio-Logic USA, Knoxville, TN, USA).

All characterization results are available in the Appendix A.

### 2.3. Experimental Procedures

The ionic compounds used in this study were synthesized according to the procedures shown in Figure 2.

Synthesis of benzyl azide: Benzyl bromide (3.00 mL, 25.3 mmol) and 30.0 mL of DMF were placed in a round-bottom flask. Sodium azide (1.81 g, 27.8 mmol) was added to the solution. The reaction mixture was stirred at room temperature overnight. The reaction mixture was added to 200 mL of water and extracted with toluene (3×5 mL). The toluene solution of benzyl azide was dried with MgSO_4_, filtered, and used directly for the following reactions. The FTIR spectra showed strong absorption of the azide group at 2094 cm^−1^. The ${}^{1}$H NMR (400 MHz, CDCl_3_) spectra showed that the mole ratio between benzyl azide (−CH_2_^−^, δ 4.34 (s, 2H)) and toluene (−CH_3_, δ 2.36 (s, 3H)) was about 1:6.

Synthesis of compound **1**: To a Schlenk flask, 4.00 g of PCl_5_ (19.2 mmol) and 60 mL of chlorobenzene were added. The reaction mixture was bubbled with N_2_ for an hour, with stirring at 0 ${}^{\circ }C$ in an ice bath. Then, 7.52 mL of N-methylcyclohexylamine (57.6 mmol) was slowly added using a constant-pressure addition funnel for 10 min. After that, 11.2 mL of triethylamine (80.7 mmol) was added using a constant-pressure addition funnel for another 10 min. The reaction mixture was stirred overnight, while the temperature increased gradually to room temperature as the ice melted. Then, 1.48 mL of propargylamine (23.2 mmol) was added to the flask. The reaction mixture was heated to 45 ${}^{\circ }C$, and stirring continued for another 24 h. The reaction mixture was poured into a beaker with 8 g of KPF_6_ and 150 mL of water. After having been stirred for about 10 min, the mixture was separated using a separatory funnel. The organic phase was saved. The water phase was extracted with cholorobenzene twice (2×10 mL). The combined organic phases were washed with water once. The solvent was evaporated using a rotary evaporator at reduced pressure and dried under vacuum; intermediate product **1’** was obtained as a brown-colored semisolid (7.42 g, 68.2% yield). ${}^{1}$H NMR (400 MHz, CDCl_3_) δ 4.47 (dt, J= 14.0, 7.1 Hz, 1H), 3.82–3.70 (m, 2H), 3.26–3.11 (m, 3H), 2.67 (d, J=10.1 Hz, 9H), 2.32 (t, J=2.4 Hz, 1H). ${}^{31}$P NMR (162 MHz, DMSO-$d_{6}$) δ 41.05 (s), −143.5 (sep, ${}^{1}J_{\mathrm{PF}}=712.8$ Hz). Both ${}^{1}$H NMR and ${}^{31}$P NMR spectra showed that the intermediate product contained impurities. However, no purification was needed at this stage. Intermediate product **1’** (7.40 g, 13.1 mmol) was dissolved in 30 mL of chlorobenzene (round-bottom flask), and 20 g of KOH water solution (50% *w*/*w*) was added. While stirring at room temperature, we added 1.87 mL of iodomethane (30.1 mmol). The reaction was stirred overnight. The mixture was poured into a beaker with 8 g of KPF_6_ and 150 mL of water. After stirring for about 10 min, we separated the mixture. The organic phase was collected. The water phase was extracted with cholorobenzene twice (2×10 mL). The combined organic phases were washed with water once. The solution was dried with MgSO_4_ overnight. After filtering, the solution was concentrated using a rotary evaporator. The product was precipitated in diethyl ether and dried under vacuum. Solid powder with a slightly yellow color was collected (3.50 g 46.2% yield). ${}^{1}$H NMR (400 MHz, DMSO-$d_{6}$) δ 3.81 (dd, J= 11.4, 2.4 Hz, 2H), 3.56 (t, J=2.3 Hz, 1H), 3.05–2.91 (m, 3H), 2.73 (d, J=9.2 Hz, 3H), 2.63 (d, J=10.6 Hz, 9H), 1.84–0.99 (m, 30H). ${}^{13}$C NMR (101 MHz, DMSO-$d_{6}$) δ 78.42, 76.30, 57.60, 54.71, 34.92, 29.61, 29.44, 25.37, 24.70. ${}^{31}$P NMR δ 44.91 (s), −143.2 (sep, ${}^{1}J_{\mathrm{PF}}=711.2$ Hz). HRMS (ESI-TOF) (m/z): [M]${}^{+}$ calculated for C_25_H_48_N_4_P^+^: 435.36111; found 435.36176. The MS spectra showed little impurity with m/z 411.36175. This impurity did not react with benzyl azide and remained intact in **1a**. Thus, the impurity could not be incorporated into the AEM.

Synthesis of compound **1a**: A piece of Cu(0) wire (d=0.25 mm, l=2 cm) was placed in a Schlenk flask. Benzyl azide solution (1.5 mL) in toluene (containing about 0.21 g of 1.6 mmol benzyl azide) was added to a round-bottom flask. Then, 5.00 mL of DMF, 0.500 g of compound **1** (0.861 mmol), 1.9 mg of CuBr_2_ (8.6 μmol), and 7.2 μL of PMDETA (34 μmol) were added to the round-bottom flask. Both flasks were degassed with N_2_ going through for 1 h. The solution was then transferred to the Schlenk flask using a syringe. The reaction mixture was stirred at 50 ${}^{\circ }C$ for 24 h. The Cu catalyst was removed by filtering through a short column with Al_2_O_3_. The solvent was slowly evaporated in the air for three days when the product crystallized. The cubic crystals were washed with Et_2_O and dried under vacuum (0.26 g, 42.3% yield). ${}^{1}$H NMR (400 MHz, DMSO-$d_{6}$) δ 8.32 (s, 1H), 7.43–7.28 (m, 5H), 5.61 (s, 2H), 4.18 (d, J=11.5 Hz, 2H), 2.81 (d, J=11.4 Hz, 3H), 2.67 (d, J=9.7 Hz, 12H), 1.84–0.91 (m, 30H). ${}^{13}$C NMR (101 MHz, DMSO-$d_{6}$) δ 142.00, 135.93, 128.71, 128.18, 127.96, 124.47, 54.73, 52.85, 42.95, 35.11, 29.59, 25.33, 24.68. ${}^{31}$P NMR δ 45.14 (s), −143.5 (sep, ${}^{1}J_{\mathrm{PF}}=711.2$ Hz). HRMS (ESI-TOF) (m/z): [M]${}^{+}$ calculated for C_32_H_55_N_7_P^+^: 568.42511; found 568.42605.

Synthesis of compound **2**: To a Schlenk flask, 4.71 g of PCl_5_ (22.6 mmol) and 60 mL of chlorobenzene were added. The reaction mixture was bubbled with N_2_ for an hour, with stirring at 0 ${}^{\circ }C$ in an ice bath. Then, 5.94 mL of morpholine (67.9 mmol) was slowly added using a constant-pressure addition funnel for 10 min. After that, 13.2 mL of triethylamine (95.0 mmol) was added using a constant-pressure addition funnel for another 10 min. The reaction mixture was stirred overnight, while the temperature increased gradually to room temperature as the ice melted. Then, 1.74 mL of propargylamine (27.1 mmol) was added to the flask. The reaction mixture was heated to 45 ${}^{\circ }C$ and continued to be stirred for another 24 h. The reaction mixture was poured into a beaker with 8 g of KPF_6_ and 150 mL of water. After stirring for about 10 min, we filtered the mixture to collect the insoluble solids. The solids were washed with *i*-PrOH and further dried under vacuum, resulting in intermediate product **2’**, white solid powder (9.39 g, 85.0% yield). ${}^{1}$H NMR (400 MHz, DMSO-$d_{6}$) δ 6.73 (dt, *J* = 13.9, 6.8 Hz, 1H), 3.83 (ddd, *J* = 14.3, 6.8, 2.5 Hz, 2H), 3.63 (m, 12H), 3.45 (t, *J* = 2.4 Hz, 1H), 3.16 (m, 12H). ${}^{31}$P NMR (162 MHz, DMSO-$d_{6}$) δ 33.28 (s), −143.5 (sep, ${}^{1}J_{\mathrm{PF}}=711.2$ Hz). Intermediate product **2’** (9.39 g, 19.2 mmol) was dissolved in 40 mL of DMSO in a round-bottom flask. Then, 1.99 mL of iodomethane (32.0 mmol) and 13.3 g of K_2_CO_3_ (96.2 mmol) were added. The mixture was stirred at room temperature overnight. The liquid was decanted into a beaker with 8 g of KPF_6_ and 150 mL of water. After stirring for about 10 min, we filtered the mixture. The solid was collected and washed with *i*-PrOH. After drying under vacuum, white solid powder was obtained (6.29 g 65.1% yield). ${}^{1}$H NMR (400 MHz, DMSO-$d_{6}$) δ 3.92 (dd, J=11.6, 2.4 Hz, 2H), 3.72–3.55 (m, 12H), 3.52 (t, J=2.3 Hz, 1H), 3.24–3.08 (m, 12H), 2.81 (d, J=9.4 Hz, 3H). ${}^{13}$C NMR (101 MHz, DMSO-$d_{6}$) δ 78.71, 76.44, 65.66, 45.13, 35.82. ${}^{31}$P NMR δ 36.61 (s), −143.5 (sep, ${}^{1}J_{\mathrm{PF}}=711.2$ Hz). HRMS (ESI-TOF) (m/z): [M]${}^{+}$ calculated for C_16_H_30_N_4_O_3_P^+^: 357.20500; found 357.20583.

Synthesis of compound **2a**: A piece of Cu(0) wire (d=0.25 mm, l=2 cm) was placed in a Schlenk flask. 1.5 mL of the benzyl azide solution in toluene (containing about 0.21 g of 1.6 mmol benzyl azide), 5.00 mL of DMSO, 0.500 g of compound **2** (1.00 mmol), 2.22 mg of CuBr_2_ (10 μmol), and 8.31 μL of PMDETA (40 μmol) were added to the round-bottom flask. Both flasks were degassed by letting N_2_ go through for 1 h. The solution was then transferred to the Schlenk flask using a syringe. The reaction mixture was stirred at 50 ${}^{\circ }C$ for 24 h. The Cu catalyst was removed by filtering through a short column with Al_2_O_3_. The solvent was slowly evaporated in the air for a week when the product crystallized. The crystals were washed with *i*-PrOH and dried under vacuum (0.32 g, 50.6% yield). ${}^{1}$H NMR (400 MHz, DMSO-$d_{6}$) δ 8.27 (s, 1H), 7.42–7.23 (m, 5H), 5.60 (s, 2H), 4.26 (d, J=11.4 Hz, 2H), 3.65–3.48 (m, 12H), 3.23–3.07 (m, 12H), 2.74 (d, J=9.3 Hz, 3H). ${}^{13}$C NMR (101 MHz, DMSO-$d_{6}$) δ 142.25, 136.01, 128.80, 128.21, 127.86, 124.28, 65.68, 52.92, 45.23, 40.42, 36.18. ${}^{31}$P NMR δ 36.81 (s), −143.4 (sep, ${}^{1}J_{\mathrm{PF}}=711.2$ Hz). HRMS (ESI-TOF) (m/z): [M]${}^{+}$ calculated for C_23_H_37_N_7_O_3_P^+^: 490.26900; found 490.27003.

Synthesis of compound **3**: A volume of 4.09 mL of N-methylpiperidine (33.6 mmol), 6.23 g of propargyl bromide solution in toluene (80% by weight, containing 42 mmol propargyl bromide), and 15 mL of hexane were added to a round-bottom flask and stirred at 60 ${}^{\circ }C$ overnight. The liquid was decanted. The insoluble part was washed with THF and dried under vacuum. A light yellow colored solid was obtained (5.08 g, 69.2% yield). ${}^{1}$H NMR (400 MHz, DMSO-$d_{6}$) δ 4.48 (d, J=2.6 Hz 2H), 4.07 (t, J=2.5 Hz 1H), 3.45–3.39 (m, 4H), 3.10 (s, 3H), 1.87–1.73 (m, 4H), 1.61–1.42 (m, 2H). ${}^{13}$C NMR (101 MHz, DMSO-$d_{6}$) δ 82.98, 72.15, 59.52, 52.36, 47.53, 20.49, 19.24.

Synthesis of compound **3a**: N-methylpiperidine (1.00 mL, 8.23 mmol) and iodomethane (0.512 mL, 8.23 mmol) were dissolved in 10 mL of acetone and stirred at room temperature overnight. The white solid powder was collected, washed with acetone, and dried under vacuum. (1.49 g, 75.1% yield) ${}^{1}$H NMR (400 MHz, DMSO-$d_{6}$) δ 3.30 (t, J=5.8 Hz, 4H), 3.05 (s, 6H), 1.83–1.71 (m, 4H), 1.51 (p, J=6.2 Hz, 2H). ${}^{13}$C NMR (101 MHz, DMSO-$d_{6}$) δ 61.51, 50.86, 20.53, 19.53.

Evaluation of the alkaline stability of model compounds **1a**, **2a**, and **3a**: KOH was dissolved in CD_3_OH to make a 2.67 M solution. Compounds **1a**, **2a**, and **3a**, 10 mg each, were dissolved in 0.200 mL of regular DMSO individually and placed in Norell Inc. 5MM, 400MHZ 7IN NMR tubes. To each sample, 0.08 μL of 2-trimethylsilylethanol was added as internal standard. Then, 0.600 mL of KOH/CD_3_OH solution was added to each sample to obtain a final KOH concentration of 2.00 M. The NMR tubes were flame-sealed and heated to 80 ${}^{\circ }C$ in an oil bath. ${}^{1}$H NMR analyses in solvent suppression mode were performed every 3 or 4 days for 30 days. Solvent suppression was applied to the −OH signal in CD_3_OH and the −CH_3_ signal in DMSO.

Synthesis of azide-modified triblock copolymer (SBS-N3).

The procedure for the synthesis of SBS-N3 is shown in Figure 3. In total, 4.00 g of SBS containing about 51.8 mmol butadiene units was dissolved in 200 mL of toluene in a round-bottom flask and stirred at 0 ${}^{\circ }C$ in an ice bath. A HBr solution in glacial acetic acid (11.2 mL containing about 64 mmol HBr) was slowly added to the flask. The reaction mixture was stirred for 24 h, while the temperature increased gradually to room temperature as the ice melted. The conversion of the double bonds was confirmed with ${}^{1}$H NMR analysis: the signal at 5.4 ppm completely disappeared. The resulting solution was washed with water 3 times to remove the majority of the acid. Then, the solution was concentrated using a rotary vaporizator. The bromide-modified polymer (SBS-Br) was obtained after precipitation in MeOH and drying under vacuum (7.67 g, 93.7% yield). SBS-Br (7.60 g containing about 48.0 mmol bromide-modified butadiene units) was dissolved in 100 mL of THF/DMF = 4/1 (*v*/*v*) in a round-bottom flask. To the solution, 0.77 g of 18-crown-6 (2.9 mmol) and 4.73 g of NaN_3_ (72.7 mmol) were added. The reaction mixture was stirred at 50 ${}^{\circ }C$ for 24 h. The conversion of the bromide groups to the azide was confirmed with ${}^{1}$H NMR analysis: the signal at 4.05 ppm completely disappeared. Insoluble solids were removed by filtering. The azide-modified polymer (SBS-N3) was precipitated in MeOH and dried under vacuum (4.60 g, 79.6% yield). The FTIR spectra showed strong absorption at 2087 cm^−1^, indicating the presence of azide groups.

General procedure for the synthesis of polyelectrolytes via click chemistry.

As shown in Figure 4, typically, a solution of SBS-N3 (0.666 g) containing about 5.54 mmol azide units in 8 mL of THF was placed in a Schlenk flask. A piece of Cu(0) wire (diameter = 0.25 mm, length = 2.00 cm) and 23.1 μL of PMDETA (0.11 mmol) were added to the flask. The reaction mixture was bubbled with N_2_ for 1 h. A total of 6.2 mg of CuBr_2_ (28 μmol) was dissolved in 2 mL of DMF and bubbled with N_2_ for 1 h. The CuBr_2_ solution was added to the Schlenk flask using a syringe. The reaction mixture was stirred at 45 ${}^{\circ }C$. Ionic compound **1**, **2**, or **3** was dissolved in 5 mL of DMF or DMSO. The selection of solvents is summarized in Table 1. After bubbling with N_2_ for 1 h, the solution of the ionic compound was slowly added to the Schlenk flask using a gas-tight syringe. The reaction was continued at 45 ${}^{\circ }C$ for 24 h. Then, 0.48 mL of propargyl alcohol (8.3 mmol), or 0.95 mL of 1-hexyne (8.3 mmol), degassed by bubbling with N_2_ for 1 h, was added to the Schlenk flask. The reaction was continued at 45 ${}^{\circ }C$ for another 24 h. The resulting solution was stored and used directly for the preparation of AEMs.

Preparation of AEMs: AEMs were prepared by drop-casting the polyelectrolyte solutions onto flat HDPE substrates. For samples SBS-c-**1**A, SBS-c-**2**A, and SBS-c-**3**A, the resulting membranes were peeled off from the substrates for analysis. However, the rest of the samples were too brittle to form stand-alone membranes or broke easily even if the membrane could be peeled off. One sample of SBS-c-1C was prepared on a glass microscope slide to evaluate the percentage of water uptake (WU%) and ion exchange capacity (IEC).

Evaluation of the IEC, WU%, and hydration number (λ): The IEC of the AEMs was determined with back titration. Typically, the AEMs were added to 0.1 M NaOH solutions and allowed to soak for 24 h. After washing thoroughly with DI water, the AEMs were added to 10.00 mL of 0.0303 M HCl solutions and allowed to soak for another 24 h. The HCl solutions were titrated with 0.0364 M NaOH. The IEC was calculated by comparing with blank samples, i.e., 10.00 mL of 0.0303 M HCl.
$$\mathrm{IEC}=\frac{\left[\begin{array}{c} \mathrm{NaOH} \end{array}\right]\times \left(V_{\mathrm{blank}}-V_{\mathrm{sample}}\right)}{m_{d}}$$
where [NaOH] is the concentration of NaOH solution, i.e., 0.0364 M, used for titration; $V_{\mathrm{blank}}$ is the volume of NaOH solution consumed by 10.00 mL of 0.0303 M HCl solution; $V_{\mathrm{blank}}$ is the volume of NaOH solution consumed by HCl solution with the sample; and $m_{d}$ is the mass of the dried sample in hydroxide form.

WU% was evaluated by comparing the wet masses and dry masses of the samples in hydroxide form.
$$\mathrm{WU}\%=\frac{m_{w}-m_{d}}{m_{d}}\times 100\%$$
where $m_{w}$ and $m_{d}$ are the wet mass and dry mass of the sample, respectively. The wet mass was weighed after dabbing off excess surface water with a dry paper towel.

The hydration number (λ) was calculated as
$$\lambda =\frac{1000\times \mathrm{WU}\%}{\mathrm{IEC}\times 18.0}$$

All measurements were performed at room temperature, i.e., 22 ${}^{\circ }C$.

Measurement of ionic conductivity of the AEMs: The AEM was cut into a rectangular shape and treated in 1 M NaCl for 24 h; then, it was allowed to exchange with hydroxide ions in 1 M NaOH solution for another 24 h. The treated samples were washed with DI water and spread out on a homemade conductivity measurement setup with four parallel platinum strips as electrodes. The in-plane resistance ($R_{\mathrm{AEM}}$) was obtained using electrochemical impedance spectroscopy (EIS). EIS was performed with a Biologic VMP3 multi-channel potentiostat in open circuit with an AC perturbation of 10 mV and a frequency range from 0.1 Hz to 50 kHz at room temperature, i.e., 22 ${}^{\circ }C$. The conductivity ($\sigma_{\mathrm{AEM}}$) was calculated using the geometries of the AEM (*l* is the distance between two voltage probes; *d* is the thickness of the membrane; and *w* is the width of the AEM).
$$\sigma_{\mathrm{AEM}}=\frac{l}{w\;\cdot \;d\;\cdot \;R_{\mathrm{AEM}}}$$

## 3. Results and Discussion

### 3.1. Synthesis of Clickable TKDAAP Compounds

The first example of a TKDAAP cation containing polyelectrolytes was synthesized in seven steps, including six steps for the synthesis of a TKDAAP-functionalized cyclooctene monomer and one step for ring-opening metathesis polymerization (ROMP) [14]. Later, a TKDAAP-functionalized styrenic monomer was synthesized using a similar strategy and polymerized via reversible addition–fragmentation chain transfer (RAFT) polymerization [15]. These long, synthetic procedures have hindered the application of TKDAAP cations in AEM development. More recently, a simpler method was developed to synthesize TKDAAP-functionalized norbornene monomers [13]. The key step of the synthesis is the reaction of PCl_5_ with 3 equiv of secondary amine, which results in a triaminochlorophosphonium intermediate. This intermediate is further treated with an excess amount of methylamine, and a TKDAAP cation with one reactive N−H group is obtained. Inspired by this work, two alkyne-functionalized TKDAAP cations were synthesized using three steps and two pot reactions (Figure 5). First, a reaction between PCl_5_ and 3 equiv of secondary amine was carried out. Then, propargylamine was added to obtain the intermediate product. Finally, the targeted TKDAAP compound was obtained via the methylation of the intermediate product under basic conditions.

After the synthesis, both compounds **1** and **2** could be precipitated in Et_2_O. Acceptable purity was achieved without further purification, which was confirmed with ${}^{1}$H NMR, ${}^{13}$C NMR, ${}^{31}$P NMR, and high-resolution mass spectroscopy analyses. In the case of compound **1**, there was a very small amount of impurity, with m/z=411.36175, presumably corresponding to a TKDAAP cation with five methyl substituents and three cyclohexyl substituents. Since this impurity would not have affected future reactions and was removed from resulting AEMs when soaking with water, it was not necessary to perform further purification. In the case of compound **2**, no detectable impurities were found. The simple synthesis and simple purification of clickable TKDAAP compounds make it possible to utilize TKDAAP cations conveniently in AEM development.

### 3.2. Alkaline Stability of Model Compounds

The reported model compounds used for the stability study (Figure 1) usually do not reflect the actual structure units of the cations in the AEMs, because the linkers between the cations and the polymers are omitted. It is necessary to synthesize the model compounds more closely by reflecting the structural units in the polymer and evaluate their stability in alkaline media. Thus, model compounds **1a** and **2a** were synthesized, and their stability was evaluated in 2 M KOH in CD_3_OH/DMSO (3/1, *v*/*v*) at 80 ${}^{\circ }C$ and compared to that of **3a** (dimethylpiperidinium; Figure 1). DMSO was added because compound **2a** was not soluble in methanol.

Dimethylpiperidinium was reported as one of the most stable QA cations, which was only slightly inferior to 6-azonia-spiro[5.5]undecane (ASU) among all QA cations investigated in a study [16]. Similar results were found in a more recent report indicating that ASU suffered about 4% degradation in 2 M KOH in methanol after 30 days and that hexylmethylpirperidinium showed about 6% degradation under identical conditions [10]. The presence of DMSO apparently accelerated the nucleophilic degradation of the cations compared with that in methanol alone. In our study, compound **3a** showed 32% degradation after 30 days, i.e., 68% remained. Using this as a reference, the stability of compounds **1a** and **2a** could be evaluated.

As expected, compound **1a** was found to be very stable with no observable changes after 30 days, i.e., >99% remained. While the excellent stability of TKDAAP cations has been extensively studied, this study further proved that (1) the triazole linker between the cation and the backbone was stable under alkaline conditions and (2) the presence of the triazole substituent did not compromise the stability of the TKDAAP cation.

Surprisingly, compound **2a** was found to be very unstable, as 50% was degraded after 1 day and over 99% was degraded after 5 days. The rationale for synthesizing compound **2** was that the presence of hydrophilic ether groups could potentially enhance the water-absorbing property of the resulting AEM, thus improving the ionic conductivity. As can be found in the following section, the hypothesis was correct, but the alkaline stability of compound **2a** was exceptionally low. In the future, it is necessary to find a way to enhance the water-absorbing property of TKDAAP-based AEMs without compromising the alkaline stability.

### 3.3. Preparation and Properties of TKDAAP-Based AEMs

The Cu(I)-catalyzed alkyne–azide cycloaddition (CuAAC) “click” reaction is used in anion exchange membrane development for both the synthesis of cation-bearing monomers and the coupling of cations with polymer backbones [17]. Using clickable TKDAAP cations, it becomes easy to evaluate TKDAAP-based AEMs with various molecular structures, thus leading to clues on how to improve the performance of the resulting AEMs. To test this idea, polystyrene-block-polybutadiene- block-polystyrene (SBS; average molecular weight of 140 K) was used as the backbone. Azide functionality was introduced in two steps of reactions. A result close to the quantitative conversion of the C=C to azide groups was achieved. The azide-modified polymers first reacted with clickable compounds **1**, **2**, and **3** in a given ratio. Then, the remaining azide groups were consumed by reacting with propargyl alcohol. Three polyelectrolytes with cations from **1**, **2**, and **3** were obtained with comparable theoretical ion exchange capacities (IECs), as shown in Figure 6 and Table 2. The polyelectrolyte solutions were cast onto HDPE substrates and dried in air to produce thin membranes. The properties of the resulting AEMs were analyzed.

According to previous reports, TKDAAP cations with methylcyclohexylamino substituents usually result in AEMs with low water-absorbing capacity and low conductivity [13,14]. This study showed the same trend: SBS-c-**1**A had 40% water uptake, and the ionic conductivity was only 1.54 mS/cm at 22 ${}^{\circ }C$. On the contrary, SBS-c-**2**A was very hydrophilic, which led to 183% water uptake and significantly enhanced conductivity, i.e., σ=13.5 ms/cm. The water uptake and conductivity of SBS-c-**2**A could be compared to those of the AEM with piperidinium cations, i.e., SBS-c-**3**A, which had a water uptake of 161% and a conductivity of 12.4 mS/cm. The study proved the hypothesis that using TKDAAP cations with more hydrophilic substituents can significantly enhance the conductivity of the resulting AEMs.

Methylcyclohexylamino-substituted TKDAAP cations are very bulky, which often leads to polyelectrolytes that are too brittle to make usable AEMs. It was found in this study that when the complementary units in SBS-c-**1**A were switched from propargyl alcohol to 1-hexyne, the resulting polyelectrolyte (SBS-c-**1**C ) had even lower water-absorbing properties and conductivity, which made it difficult to obtain any measurable values. SBS-c-**1**C was also so brittle that it could not form stand-alone membranes. Similarly, polyelectrolyte SBS-c-**1**B with propargyl alcohol as the complementary unit and with doubled the cation content of SBS-c-**1**A was also too brittle to make usable membranes.

In the case of cation **2**, doubling the cation content also slightly increased the rigidness of the resulting polymer, i.e., SBS-c-**2**B. This polyelectrolyte could still form membranes while maintaining its integrity after being peeled off from the substrate. However, the membranes swell too much when immersed in water and became very weak.

As illustrated in this study, it was very convenient to use clickable TKDAAP compounds to synthesize polyelectrolytes with different cations and complementary units. The cation content could be changed easily to test and compare the properties of the resulting polyelectrolytes. Potentially, it is also possible to design different polymer backbones to achieve AEMs with better performance.

## 4. Conclusions

Clickable TKDAAP compounds were synthesized with a simple approach. It is easy to incorporate different clickable cations into various polymer backbones with adjustable cation content. As a result, a systematical study on TKDAAP-based AEMs becomes feasible. From this study, it can be concluded that the triazole linkers resulting from a CuAAC “click” reaction are stable under alkaline conditions. However, TKDAAP cations with different substituents can exhibit dramatically different alkaline stability. Thus, the stability of each TKDAAP cation to be used for AEMs needs to be evaluated individually. To enhance the conductivity of TKDAAP-based AEMs, it is necessary to increase the hydrophilicity of the TKDAAP-containing polyelectrolytes. In addition, ductile and non-swelling components should be introduced to the polyelectrolyte in order to mitigate brittleness in polymers and limit the level of water uptake.

## Figures and Tables

**Scheme 1 polymers-15-00352-sch001:**
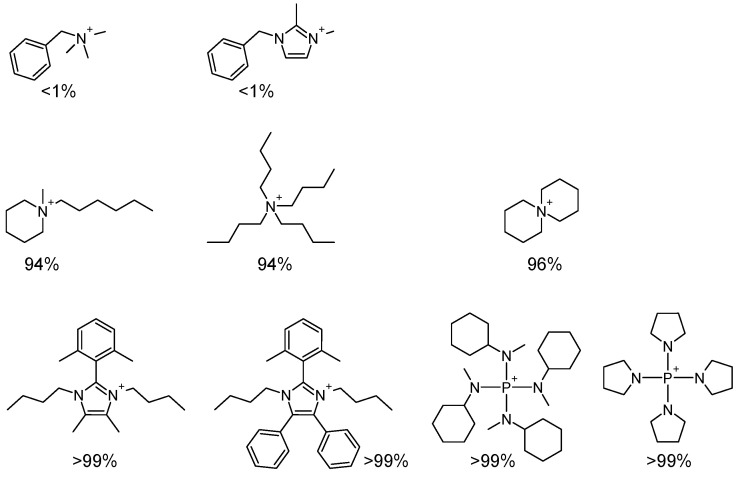
Examples of model compound stability characterized as the percentage remaining after 30 days at 80 ${}^{\circ }C$ in 2 M KOH in CD_3_OH.

**Scheme 2 polymers-15-00352-sch002:**
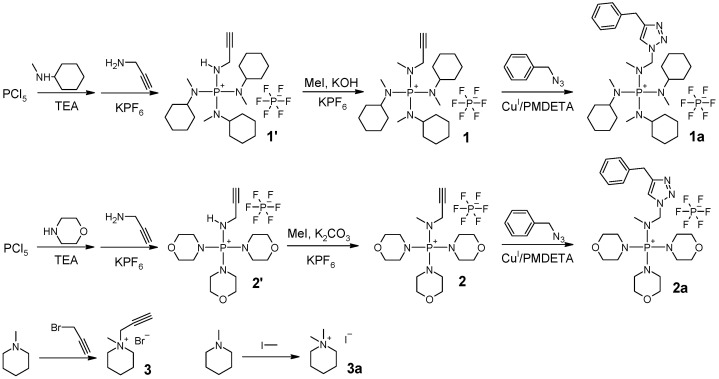
Synthesis of the ionic compounds used in this study.

**Scheme 3 polymers-15-00352-sch003:**
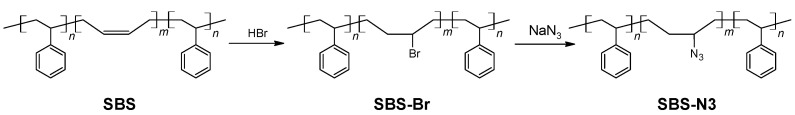
Synthesis of azide-modified triblock copolymer.

**Scheme 4 polymers-15-00352-sch004:**
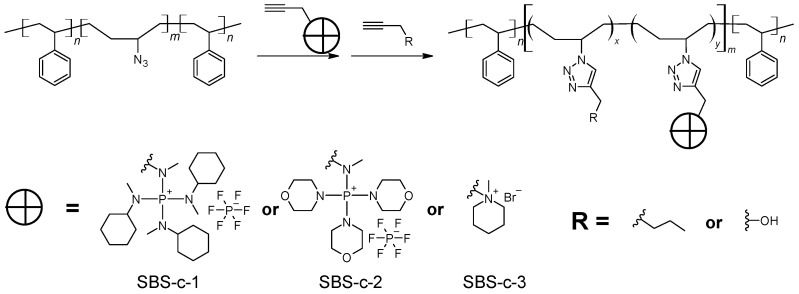
Synthesis of the polyelectrolytes via click chemistry.

**Scheme 5 polymers-15-00352-sch005:**
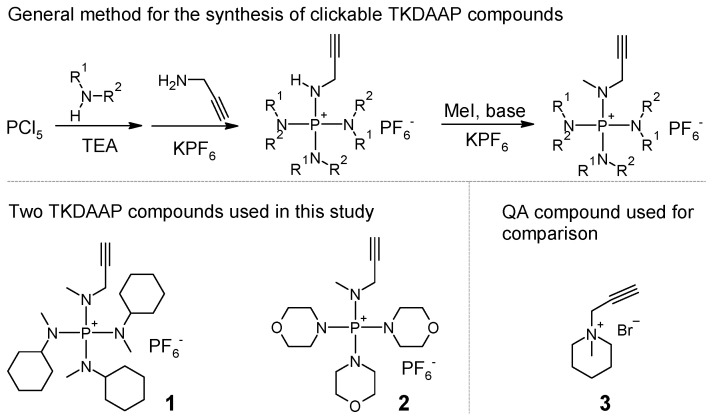
Ionic compounds used in this study.

**Figure 1 polymers-15-00352-f001:**
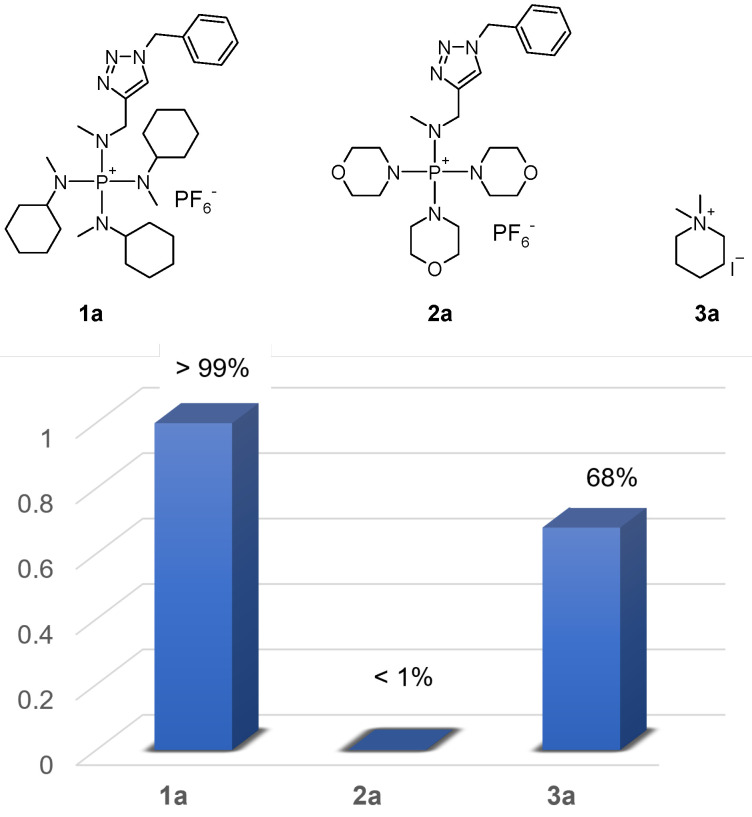
Percentage remaining of model compounds in 2 M KOH in CD_3_OH/DMSO (3/1, *v*/*v*) at 80 ${}^{\circ }C$ after 30 days.

**Scheme 6 polymers-15-00352-sch006:**
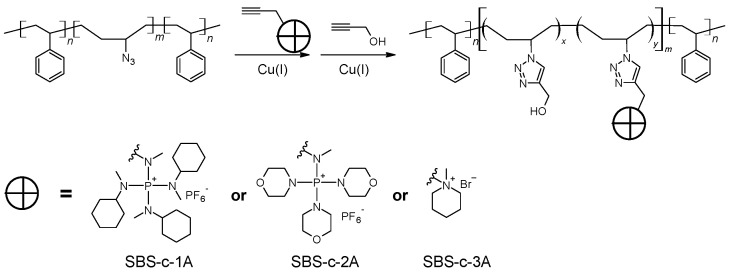
Polyelectrolytes synthesized via click chemistry.

**Table 1 polymers-15-00352-t001:** Ratios of reagents for the synthesis of polyelectrolytes.

Sample Name	Compound	Mass (g)	Moles (mmol)	Alkyne	Solvent
SBS-c-**1**A	**1**	0.755	1.30	propargyl alcohol	DMF
SBS-c-**1**B	**1**	1.49	2.57	propargyl alcohol	DMF
SBS-c-**1**C	**1**	0.755	1.30	1-hexyne	DMF
SBS-c-**2**A	**2**	0.668	1.33	propargyl alcohol	DMSO
SBS-c-**2**B	**2**	1.33	2.65	propargyl alcohol	DMSO
SBS-c-**3**A	**3**	0.242	1.11	propargyl alcohol	DMF

**Table 2 polymers-15-00352-t002:** Properties of the AEMs obtained in this study.

Sample	IEC${}_{{\mathrm{th}}^{a}}$	IEC${}_{{\mathrm{ex}}^{b}}$	WU%^*c*^	$\lambda^{d}$	$\sigma^{e}$ (mS/cm)
SBS-c-**1**A	0.87	0.77±0.01	40%±8%	29	1.54
SBS-c-**2**A	0.95	1.06±0.01	183%±4%	96	13.5
SBS-c-**3**A	1.02	1.52±0.02	161%±3%	59	12.4

*a* Theoretical IEC values calculated according to the amount of the molar ratio of reagents used during the
synthesis of the AEMs. *b* IEC values experimentally determined via back titration. *c* Percentage of water uptake in the hydroxide form. d Hydration number. e Ionic conductivity measured in the hydroxide form. All measurements
were conducted at room temperature, i.e., 22 °C.

## Data Availability

The data presented in this study are all available within this article and the Appendix A.

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
