# Peer review of "Efficient Synthesis of High-Performance Anion Exchange Membranes by Applying Clickable Tetrakis(dialkylamino)phosphonium Cations"

_polymers, 2023, doi:10.3390/polym15020352_

Round 1

Reviewer 1 Report

The manuscript "Clickable tetrakis(dialkylamino)phosphonium cations for efficient anion exchange membrane development" proposes the usage of clickable TKDAAP compounds into polymer backbones coupled with different cations to be used in anion exchange membrane fuel cells. The manuscript is well presented and the aspects proposed by the authors are well presented. Personally, I had some difficulties connecting the important section of the introduction that the Authors dedicate to the importance of stability to the results. I mean that probably a deeper investigation of the degradation process and how the key parameters (like conductivity) evolve during the ageing test should be appreciable. I found that this aspect has a low profile in the paper and more evidence is dedicated to the material synthesis procedure. Nevertheless, research is not made my personal comment, and the paper is well written and can be considered to be published in Polymers. Only two minor changes are suggested.

LINE20 - "due to improved oxygen reduction reaction (ORR) kinetics". This sentence does not explain why the ORR kinetic of AEMFC is improved compared with PEMFC

Figure 1 appears before the reference text. Please reorganize the page.

Author Response

Thanks for the comments. Indeed, the synthesis and the test of alkaline stable membranes are very time-consuming, which is why a highly efficient method is needed. This article mainly reports the proof of the new synthetic procedure. Using this synthetic method we are able to test a large variety of AEMs with different TKDAAP cations and various polymer backbones and the continuous study will be reported separately. 

Concerning the reviewer's two suggestions

 LINE20 - "due to improved oxygen reduction reaction (ORR) kinetics". This sentence does not explain why the ORR kinetic of AEMFC is improved compared with PEMFC

The ORR kinetics under acidic and basic conditions were thoroughly studied and discussed in the reference articles. To be concise, it is not explicitly explained here. 

Figure 1 appears before the reference text. Please reorganize the page.

The manuscript has been reorganized. 

Reviewer 2 Report

I greatly enjoyed this manuscript and found the work to be thorough and compelling.

Author Response

Thanks for the encouraging comments. 

Reviewer 3 Report

Title: Clickable tetrakis(dialkylamino)phosphonium cations for efficient anion exchange membrane development

Recommendation: Rejected needed as noted.

There are a few ambiguous aspects that the writers ought to make an effort to clarify.

1. Is there a published article that is similar in the literature?

2. In the introduction section, it must be made apparent how the present work is innovative. The results of the investigation are not clearly explained.

3. Place a reference before end of the sentence.

4. Designing the polymer backbones and cations in a single step is preferable than designing them individually and combining them in various ways. What are the advantages of using this procedure?

5. Why is the synthesis process so complicated and multi-step?

6. Poor English writing, the document has to be carefully revised because it contains several grammatical, structural, and typing errors.

7. Unable to mention the units since there is no single style or standard for doing so.

8. Maybe accepted after careful revision of the manuscript.  

Author Response

Thanks for the comments. Please see follows the answers to the questions.

1. Is there a published article that is similar in the literature?

We have cited previous reports on TKDAAP cation based AEMs and the application of click chemistry in AEM development. Those are the closest reports. The synthesis of clickable TKDAAP has never been reported to the best of our knowledge. 

2. In the introduction section, it must be made apparent how the present work is innovative. The results of the investigation are not clearly explained.

The main innovation is that we introduced a new method to utilize TKDAAP cations in AEM synthesis, which was otherwise very challenging. We have revised the introduction part to better explain the motivation of this study. The explanation can be found in the second and third paragraphs. 

3. Place a reference before end of the sentence.

Thanks for the suggestions. The manuscript has been revised accordingly. 

4. Designing the polymer backbones and cations in a single step is preferable than designing them individually and combining them in various ways. What are the advantages of using this procedure?

As has been discussed in the manuscript, the synthesis of TKDAAP based AEMs usually takes many steps. However, there is no guarantee that the product would have proper physical and chemical properties. If the performance is not as good as expected, it is necessary to perform the whole synthetic procedure again varying the components and the ratio of the components. The synthesis and tests are very time-consuming. By using clickable TKDAAP cations, it becomes possible to synthesize the cation once and use it for various AEM formulas. Combining the cation with the polymer backbone is a simple one-step reaction, i.e., click chemistry. 

5. Why is the synthesis process so complicated and multi-step?

The synthesis of functional TKDAAP cations is indeed complicated. But they are also very valuable because of their excellent alkaline stability. We have successfully reduced the number of synthetic steps to two to get the clickable cation. Then we could use the cation with different backbones and vary the fraction of the cation easily. 

6. Poor English writing, the document has to be carefully revised because it contains several grammatical, structural, and typing errors.

We have asked a native speaker to check the language style thoroughly. 

7. Unable to mention the units since there is no single style or standard for doing so.

Sorry, I could not quite understand the question. 

8. Maybe accepted after careful revision of the manuscript.

Thanks, we have made careful revisions.  

Round 2

Reviewer 3 Report

Much better